# Lattice random walks and quantum A-period conjecture

**Li Gan**

Galileo Galilei Institute for Theoretical Physics, INFN, 50125 Firenze, Italy

li.gan92@gmail.com

## Abstract

We derive explicit closed-form expressions for the generating function $C_N(A)$, which enumerates classical closed random walks on square and triangular lattices with $N$ steps and a signed area $A$, characterized by the number of moves in each hopping direction. This enumeration problem is mapped to the trace of powers of anisotropic Hofstadter-like Hamiltonian and is connected to the cluster coefficients of exclusion particles: Exclusion strength parameter $g = 2$ for square lattice walks, and a mixture of $g = 1$ and $g = 2$ for triangular lattice walks. By leveraging the intrinsic link between the Hofstadter model and high energy physics, we propose a conjecture connecting the above signed area enumeration $C_N(A)$ in statistical mechanics to the quantum A-period of associated toric Calabi–Yau threefold in topological string theory: Square lattice walks correspond to local $\mathbb{F}_0$ geometry, while triangular lattice walks are associated with local $\mathcal{B}_3$.


## Contents

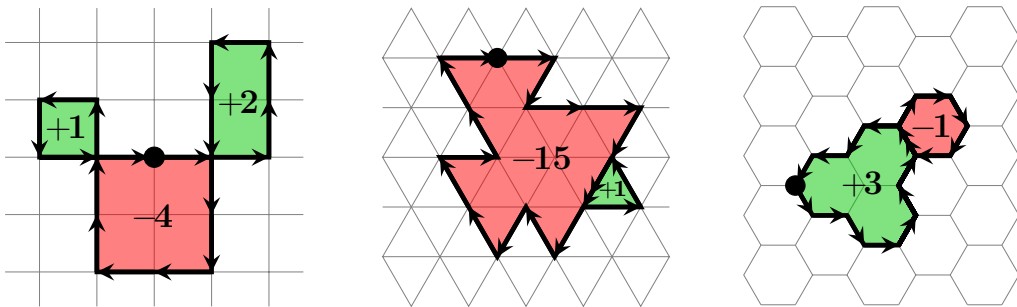

Figure 1: A closed random walk of length 18 on the square (left), triangular (middle), and honeycomb lattice (right), starting and ending at the bullet point, with signed area $-1$, $-14$ and $2$, respectively. The region inside the walk, i.e., winding sector, is colored green if its area is positive, otherwise it is colored red.

## 1 Introduction

After half a century, the Hofstadter model [1] remains a vibrant subject of research in condensed matter physics and beyond. It describes the motion of a charged particle hopping on a square lattice in the presence of a perpendicular magnetic field. Its energy spectrum, known as the "Hofstadter butterfly," showcases fractal structures that have intrigued both mathematicians (see, e.g., [2,3]) and physicists, especially in the contexts of the quantum Hall effect [4,5] and topological quantum numbers [6]. The Hofstadter butterfly has been experimentally observed [7], and similar butterfly structures have been extensively studied on various lattices, including 2D lattices such as the triangular lattice [8–11], honeycomb lattice [12–14], and kagome lattice [15,16]; 3D lattices such as the cubic lattice [17–20], tetragonal monoatomic and double-atomic lattices [21]; 4D lattices [22]; and non-Euclidean hyperbolic lattices [23].

In statistical physics, the Hofstadter model is closely related to classical lattice random walks, where counting problems related to the signed area [24] (also known as the algebraic area) of closed lattice random walks in 2D can be mapped [25] onto to moments of the Hofstadter Hamiltonian. Recent advances have greatly improved our ability to solve these problems, yielding closed-form expressions for $C_N(A)$, which enumerate the number of closed lattice walks with $N$ steps and a signed area $A$ on various lattice types, such as square [26,27], triangular (both standard [28] and chiral walks [27,29]), and honeycomb [30]. The signed area of a planar closed walk is defined as the area enclosed by the walk, weighted by the winding number in each winding sector. By convention, the area is positive if the walk moves counterclockwise around the winding sector (see Figure 1). Significant connections have been established between the signed area enumeration, exclusion statistics, and combinatorics of generalized Dyck and Motzkin paths. Exclusion statistics, originally proposed in [31,32], generalize Bose–Einstein statistics and Fermi–Dirac statistics, and play an important role in these connections. As a result, the signed area enumeration is no longer a purely mathematical problem but is now closely related to the Hofstadter model and statistical mechanics.

Additionally, recent studies [33–37] in high energy physics have revealed a profound connection between the Hofstadter model and toric Calabi–Yau threefolds in topological string theory. Specifically, the Hofstadter model on a square lattice corresponds to local $\mathbb{F}_0$ geometry (i.e., local $\mathbb{P}^1 \times \mathbb{P}^1$), the triangular lattice corresponds to local $\mathcal{B}_3$ (i.e., three-point blow-up of the local $\mathbb{P}^2$), and the honeycomb lattice also corresponds to local $\mathcal{B}_3$, albeit with different complex moduli parameters. These connections tie the model to a range of fields, including Chern–Simons theory, mirror symmetry, enumerative geometry, integrable systems, and random matrix theory [38–44]. An immediate question arises: Given the relation between the

Hofstadter model, lattice random walks, and topological string theory, is there a direct correspondence that links quantities in lattice random walks to those in topological string theory? If such a correspondence exists, it would open the door to applying statistical mechanics methods to investigate toric Calabi–Yau geometries and their spectra, while also enabling techniques from topological string theory to shed light on lattice models. In this paper, we present an initial conjectural response to this intriguing question.

In this paper, we address two related aspects of lattice random walks. First, in Section 2, we tackle the enumeration problem for lattice random walks, deriving granular results, where $C_N(A)$ also accounts for the number of moves in each direction, regardless of their sequence. By calculating the trace of powers of *anisotropic* Hofstadter-like Hamiltonians, we derive counting formulae for square and triangular lattice walks and show their connection to exclusion statistics through Kreft coefficients obtained from the secular determinant. Second, in Section 3, we propose a conjecture based on the shared link between lattice random walks and topological string theory through the Hofstadter model. This conjecture posits a direct relation between the refined enumeration counts and the quantum A-period of the corresponding toric Calabi–Yau threefold, which we validate using established results from the literature.

## 2 Signed area enumeration of lattice random walks

In this section, we first review the connection between signed area enumeration and the Hofstadter model. We then focus on the specific cases of square and triangular lattices and calculate their signed area enumeration.

### 2.1 Hofstadter-like Hamiltonians

For square lattice walks, we introduce the four lattice hopping operators $u$, $u^{-1}$, $v$, and $v^{-1}$, corresponding to moves to the right, left, up, and down, respectively. These operators satisfy the commutation relation

$$v\,u = Q\,u\,v\,, \tag{1}$$

which amounts to saying that the 4-step closed walk "right-up-left-down" around a unit lattice cell in a counterclockwise direction encloses an area 1, i.e., $v^{-1}u^{-1}vu = Q^1$ (writing the hopping operators from right to left). Here, the variable $Q$ is a placeholder for the signed area $A$ and commutes with all the hopping operators. The signed area $A$ enclosed by a walk can thus be computed by reducing the corresponding hopping operators to $Q^A$ using the relation (1). To count moves in each direction, we introduce variables to the operators: $a$ and $a'$ for $u$ and $u^{-1}$, and $b$ and $b'$ for $v$ and $v^{-1}$. These variables commute with

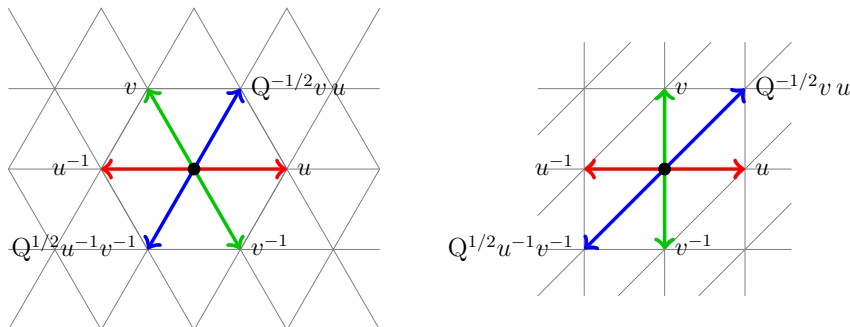

Figure 2: Six hopping operators on the standard (left) and deformed (right) triangular lattice.

each other and serve as placeholders to track the walk. For example, the walk that goes right-left-left-right corresponds to $(au)(a'u^{-1})(a'u^{-1})(au) = a^2 a'^2$, showing two moves to the right and two to the left. As a result, the $u$- and $v$-independent part of the expansion $(au + a'u^{-1} + bv + b'v^{-1})^N = \sum_A C_N(A)Q^A + \cdots$ yields $C_N(A)$, which enumerates closed square lattice walks of (necessarily even) length $N$ enclosing a signed area $A$ and specifies the moves in each direction.

A similar approach can be extended to random walks on a triangular lattice (see Figure 2, left). Considering the connection between the deformed triangular lattice (see Figure 2, right) and the square lattice, we define the area of a unit triangular lattice cell as $1/2$. Consequently, the signed area $A$ of triangular lattice walks takes half-integer values: $0$, $\pm 1/2$, $\pm 1$, ... In addition to the four hopping operators $u$, $u^{-1}$, $v$, and $v^{-1}$, along with their corresponding variables $a$, $a'$, $b$, and $b'$ as in the square case, we introduce two additional hopping operators $Q^{-1/2}vu$ and $Q^{1/2}u^{-1}v^{-1}$. These are associated with the variables $c$ and $c'$, respectively, to account for diagonal moves "lower-left" and "upper-right." Similarly, the $u$- and $v$-independent part in $(au + a'u^{-1} + bv + b'v^{-1} + cQ^{1/2}u^{-1}v^{-1} + c'Q^{-1/2}vu)^N = \sum_A C_N(A)Q^A + \cdots$ yields $C_N(A)$.[1]

By interpreting $Q = e^{2\pi i\Phi/\Phi_0}$, where $\Phi$ is the flux of an external magnetic field through a unit lattice cell and $\Phi_0$ the flux quantum, the operator

$$H_{\mathrm{tri}} = au + a'u^{-1} + bv + b'v^{-1} + cQ^{1/2}u^{-1}v^{-1} + c'Q^{-1/2}vu \tag{2}$$

becomes the Hamiltonian of a charged particle hopping on a triangular lattice in the presence of a perpendicular magnetic field, with $a, a', b, b', c, c' \in \mathbb{R}^+$ representing general transition amplitudes. Similar to the standard Hofstadter model on the square lattice, the Hamiltonian $H_{\mathrm{tri}}$ describes the Hofstadter model on a triangular lattice, with its spectrum analyzed in [8–10].

When the magnetic flux is rational, $Q = e^{2\pi ip/q}$ with $p, q$ coprime integers, the lattice hopping operators $u$ and $v$ can be represented by the $q \times q$ "clock" and "shift" matrices

$$u = e^{ik_x}\begin{pmatrix} Q & 0 & 0 & \cdots & 0 & 0 \\ 0 & Q^2 & 0 & \cdots & 0 & 0 \\ 0 & 0 & Q^3 & \cdots & 0 & 0 \\ \vdots & \vdots & \vdots & \ddots & \vdots & \vdots \\ 0 & 0 & 0 & \cdots & Q^{q-1} & 0 \\ 0 & 0 & 0 & \cdots & 0 & 1 \end{pmatrix}, \quad v = e^{ik_y}\begin{pmatrix} 0 & 1 & 0 & \cdots & 0 & 0 \\ 0 & 0 & 1 & \cdots & 0 & 0 \\ 0 & 0 & 0 & \cdots & 0 & 0 \\ \vdots & \vdots & \vdots & \ddots & \vdots & \vdots \\ 0 & 0 & 0 & \cdots & 0 & 1 \\ 1 & 0 & 0 & \cdots & 0 & 0 \end{pmatrix}, \tag{3}$$

where $k_x$ and $k_y$ are quasimomenta in the $x$ and $y$ directions. Thus, in the quantum world, selecting the $u$- and $v$-independent part of $H_{\mathrm{tri}}^N$ translates into computing the "full" trace of $H_{\mathrm{tri}}^N$ defined by

$$\mathbf{Tr}\,H_{\mathrm{tri}}^N = \frac{1}{q}\int_0^{2\pi}\int_0^{2\pi}\frac{dk_x}{2\pi}\frac{dk_y}{2\pi}\,\mathrm{tr}\,H_{\mathrm{tri}}^N,$$

where tr denotes the matrix trace. Due to the fact that $\mathbf{Tr}\,u^m v^n = \delta_{m,0}\delta_{n,0}$, only terms with an equal number of $u$ and $u^{-1}$, $v$ and $v^{-1}$ survive, corresponding to all closed walks. The integration over $k_x$ and $k_y$ eliminates the unwanted terms containing $u^{qm}$ and $v^{qn}$ with $m, n \neq 0$, which correspond to open walks but can be closed by $q$-periodicity. Using the full trace, we obtain the full generating function for the signed area enumeration, i.e.,

$$\mathbf{Tr}\,H_{\mathrm{tri}}^N = \sum_A C_N(A)Q^A. \tag{4}$$

---

[1]Since this paper concerns only closed walks, certain variables introduced for both square and triangular lattice walks are redundant due to the closure-condition constraint. The number of variables can be reduced, for example, by fixing $a = b = 1$ in both cases, without loss of generality. Nevertheless, we retain all variables to keep step counts in each direction explicit, facilitating later discussions and making our results more transparent and easier to interpret. We thank an anonymous referee for pointing out this simplification.

Therefore, this counting problem is related to the Hofstadter model and is reformulated as calculating the full trace $\mathbf{Tr}\,H_{\mathrm{tri}}^N$. In prior work, two general approaches have been proposed. The first approach relies on the computation of the secular determinant $\det(I - zH')$ of an equivalent Hamiltonian $H'$, which yields the Kreft coefficients [45] and reveals connections to exclusion statistics [27]. The second approach is to directly calculate the matrix trace, $\mathrm{tr}\,H'^N$, which can be mapped to the combinatorics of generalized periodic Dyck or Motzkin paths [46]. Both approaches yield the same result. In the following subsections, we will extend the first approach to anisotropic Hamiltonians and derive the signed area enumeration $C_N(A)$ of both square and triangular lattice walks.

## 2.2   Signed area enumeration of square lattice walks

Before analyzing the triangular lattice, we first consider the square lattice as a special case by setting $a = a = 0'$ in (2). Although $a$ and $a'$ are positive in the general formulation, this simplification corresponds to random walks on a deformed square lattice. By symmetry, setting $b = b' = 0$ or $c = c' = 0$ yields equivalent enumeration results.[2]

For the square lattice case, the Hamiltonian

$$H_{\mathrm{sq}} = bv + b'v^{-1} + c\,Q^{1/2}u^{-1}v^{-1} + c'Q^{-1/2}vu = \begin{pmatrix} 0 & f_1 & 0 & \cdots & 0 & g_q \\ g_1 & 0 & f_2 & \cdots & 0 & 0 \\ 0 & g_2 & 0 & \cdots & 0 & 0 \\ \vdots & \vdots & \vdots & \ddots & \vdots & \vdots \\ 0 & 0 & 0 & \cdots & 0 & f_{q-1} \\ f_q & 0 & 0 & \cdots & g_{q-1} & 0 \end{pmatrix},$$

is written as a $q \times q$ matrix, where $f_k = (b + c'\,\mathrm{e}^{ik_x}Q^{k+\frac{1}{2}})\mathrm{e}^{ik_y}$ and $g_k = (b' + c\,\mathrm{e}^{-ik_x}Q^{-k-\frac{1}{2}})\mathrm{e}^{-ik_y}$. Its spectrum follows from the zeros of the secular determinant

$$\det(I - zH_{\mathrm{sq}}) = \sum_{n=0}^{\lfloor q/2 \rfloor} (-1)^n Z_n z^{2n} - \left( \prod_{k=1}^{q} f_k + \prod_{k=1}^{q} g_k \right) z^q,$$

where $\lfloor \cdot \rfloor$ denotes the floor function. As we shall see, the $k_x$- and $k_y$-independent coefficient $Z_n$, known as the Kreft coefficient, is at the core of the lattice walks signed area enumeration. To determine $Z_n$, we need to eliminate the spurious "Wilson loop" contribution term $-(\prod_{k=1}^{q} f_k + \prod_{k=1}^{q} g_k)z^q$. This term arises from the effects of momentum periodicity on the Hofstadter model and will disappear if the upper right and lower left corners of $H$ vanish. To achieve this, we set $\mathrm{e}^{ik_x} = -\frac{b}{c'}Q^{-1/2}$ and $k_y = 0$, and neglect $g_q$ by treating it as zero. Denote this matrix $H_2$, where the subdiagonal elements become

$$f_k = b(1 - Q^k), \qquad g_k = b' - \frac{cc'}{b}Q^{-k}.$$

Its secular determinant $\det(I - zH_2) = \sum_{n=0}^{\lfloor q/2 \rfloor}(-1)^n Z_n z^{2n}$ do not have the spurious term anymore. Therefore, $H_2$ can be regarded as an equivalent Hofstadter Hamiltonian in the sense that its matrix trace $\mathrm{tr}\,H_2^N$ equals the full trace $\mathbf{Tr}\,H_{\mathrm{sq}}^N$ up to a scaling factor:

$$\mathbf{Tr}\,H_{\mathrm{sq}}^N = \frac{1}{q}\mathrm{tr}\,H_2^N, \text{ for } N < q.$$

---

[2]By setting $c = c' = 0$, the Hamiltonian becomes the standard Hofstadter one. The transformation $u \to -vu, v \to v$, which leaves their commutation relation invariant, is necessary to vanish the spurious term in the secular determinant. See [27]. This is essentially equivalent to directly considering the random walks on a deformed square lattice, as will be shown in this subsection.

For $N \geq q$, $\frac{1}{q}\mathrm{tr}\, H_2^N$ includes additional terms corresponding to open walks, which, as mentioned in Section 1, are treated as closed due to the $q$-periodicity, and is therefore no longer equal to $\mathbf{Tr}\, H_{\mathrm{sq}}^N$.

The recursion arising from the expansion of $\det(I - zH_2)$ yields the Kreft coefficient, as derived in Appendix 1 of Ref. [30]:

$$Z_0 = 1\,, \qquad Z_n = \sum_{k_1=1}^{q-2n+1}\sum_{k_2=1}^{k_1}\cdots\sum_{k_n=1}^{k_{n-1}} s_{k_1+2n-2}s_{k_2+2n-4}\cdots s_{k_{n-1}+2}s_{k_n} = \sum_{\substack{k_{i+1}-k_i\geq 2 \\ k_1\geq 1,\ k_n\leq q-1}} s_{k_1}s_{k_2}\cdots s_{k_n}\,,$$

with the spectral function

$$s_k = f_k\, g_k = (1-Q^{-k})(cc' - bb'Q^k)\,.$$

In statistical mechanics, $Z_n$ can be interpreted as the $n$-body partition function for $n$ particles with one-body spectrum $\epsilon_k$ with Boltzmann factor $e^{-\beta\epsilon_k} = s_k$. The $+2$ shifts indicate that these particles obey $g = 2$ exclusion statistics ($g = 0$ for bosons, $g = 1$ for fermions, $g \geq 2$ for stronger exclusion than fermions), i.e., no two particles can occupy two adjacent quantum states. These particles are thus referred to as "exclusons."

By introducing the cluster coefficients $b_n$ via $\log\left(\sum_{n=0}^{\lfloor q/2\rfloor} Z_n x^n\right) = \sum_{n=1}^{\infty} b_n x^n$ with fugacity $x = -z^2$ and using the identity $\log\det(I - zH_2) = \mathrm{tr}\log(I - zH_2) = -\sum_{N=1}^{\infty}\frac{z^N}{N}\mathrm{tr}\, H_2^N$, we establish a connection between the full generating function for signed area enumeration of square lattice walks and the cluster coefficient $b_n$ associated with $g = 2$ exclusion statistics, that is, for $N < q$,

$$\mathbf{Tr}\, H_{\mathrm{sq}}^N = \frac{1}{q}\mathrm{tr}\, H_2^N = N(-1)^{n+1}\frac{1}{q}b_n = N\sum_{\substack{l_1,l_2,\ldots,l_j \\ \text{composition of } N/2}} c_2(l_1,l_2,\ldots,l_j)\frac{1}{q}\sum_{k=1}^{q-j} s_k^{l_1}s_{k+1}^{l_2}\cdots s_{k+j-1}^{l_j}\,, \quad (5)$$

where $N = 2n$ and the combinatorial coefficients

$$c_2(l_1,l_2,\ldots,l_j) = \frac{1}{l_1}\prod_{i=2}^{j}\binom{l_{i-1}+l_i-1}{l_i}$$

are labeled by the compositions $l_1,l_2,\ldots,l_j$ of $n$, i.e., ordered partitions of $n$. See [46] for a combinatorial interpretation of $c_2(l_1,l_2,\ldots,l_j)$ in terms of counting periodic Dyck paths.

To compute the trigonometric sum $\frac{1}{q}\sum_{k=1}^{q-j} s_k^{l_1}s_{k+1}^{l_2}\cdots s_{k+j-1}^{l_j}$ in (5), we use the isotropic Hofstadter Hamiltonian result [26, 29], where the standard spectral function reads

$$S_k = s_k|_{b=b'=c=c'=1} = (1-Q^{-k})(1-Q^k) = 4\sin^2(k\pi p/q)\,,$$

leading to

$$\frac{1}{q}\sum_{k=1}^{q-j} S_k^{l_1}S_{k+1}^{l_2}\cdots S_{k+j-1}^{l_j} = \sum_{A=-\lfloor(l_1+l_2+\cdots+l_j)^2/4\rfloor}^{\lfloor(l_1+l_2+\cdots+l_j)^2/4\rfloor} Q^A \sum_{k_3=-l_3}^{l_3}\sum_{k_4=-l_4}^{l_4}\cdots\sum_{k_j=-l_j}^{l_j}\binom{2l_1}{l_1+A+\sum_{i=3}^{j}(i-2)k_i}$$

$$\times\binom{2l_2}{l_2-A-\sum_{i=3}^{j}(i-1)k_i}\prod_{i=3}^{j}\binom{2l_i}{l_i+k_i}\,.$$

Replacing all binomials of the form $\binom{2l}{k}$ with $\sum_{m=0}^{l}\binom{l}{m}\binom{l}{k-m}(bb')^m(cc')^{l-m}$ for the anisotropic case, we obtain the generating function

$$
\begin{aligned}
C_N(A) = N \sum_{\substack{l_1,l_2,\ldots,l_j \\ \text{composition of } N/2}} \frac{1}{l_1} \prod_{i=2}^{j} \binom{l_{i-1}+l_i-1}{l_i} \sum_{k_3=-l_3}^{l_3} \sum_{k_4=-l_4}^{l_4} \cdots \sum_{k_j=-l_j}^{l_j} & \left[ \sum_{m=0}^{l_1} \binom{l_1}{m}\binom{l_1}{l_1+A+\sum_{i=3}^{j}(i-2)k_i-m} \right. \\
\times (bb')^m(cc')^{l_1-m} \left[ \sum_{m=0}^{l_2} \binom{l_2}{m}\binom{l_2}{l_2-A-\sum_{i=3}^{j}(i-1)k_i-m} (bb')^m(cc')^{l_2-m} \right] & \left. \prod_{i=3}^{j} \sum_{m=0}^{l_i} \binom{l_i}{m} \right. \\
\times \binom{l_i}{l_i+k_i-m}(bb')^m(cc')^{l_i-m}, &
\end{aligned}
\tag{6}
$$

for the enumeration of square lattice walks with $N$ steps and signed area $A$ (bounded by $\lfloor N^2/16 \rfloor$), characterized by the number of moves in the four directions ($b, b'$ for right and left, $c, c'$ for up and down). At first glance, the explicit expression for $C_N(A)$ in (6) may seem complicated. However, its three-part structure provides a clear interpretation: The summation indexed by the composition, the combinatorial factor that counts specific configurations, and the nested multiple sum that corresponds to the single trigonometric sum at the core of the cluster coefficients. Note that the special case $(b, b', c, c') = (1, 1, \lambda, \lambda)$ was also obtained in [47].

## 2.3 Signed area enumeration of triangular lattice walks

For general $a, a', b, b', c, c' \in \mathbb{R}^+$, the Hamiltonian

$$
H_{\text{tri}} = \begin{pmatrix}
\tilde{s}_1 & f_1 & 0 & \cdots & 0 & g_q \\
g_1 & \tilde{s}_2 & f_2 & \cdots & 0 & 0 \\
0 & g_2 & \tilde{s}_3 & \cdots & 0 & 0 \\
\vdots & \vdots & \vdots & \ddots & \vdots & \vdots \\
0 & 0 & 0 & \cdots & \tilde{s}_{q-1} & f_{q-1} \\
f_q & 0 & 0 & \cdots & g_{q-1} & \tilde{s}_q
\end{pmatrix}
$$

becomes a tridiagonal matrix with nonvanishing corners. The main diagonal elements, arising from the operators $u$ and $u^{-1}$, read $\tilde{s}_k = a\, e^{ik_x}Q^k + a'\, e^{-ik_x}Q^{-k}$. We follow the procedure in Section 2.2, setting $e^{ik_x} = -\frac{b}{c'}Q^{-1/2}$ and $k_y = 0$, and neglecting $g_q$, so that both corners vanish. Denote this matrix as $H_{1,2}$. We then have

$$
\det(I - zH_{1,2}) = \sum_{n=0}^{q} (-z)^n Z_n \,,
$$

with the Kreft coefficient

$$
Z_0 = 1 \,, \qquad Z_n = \sum_{\substack{g_1+\cdots+g_j=n \\ g_i \in \{1,2\}}} \sum_{\substack{k_{i+1}-k_i \geq g_i \\ k_1 \geq 1,\ k_j \leq q-g_j+1}} s_{k_1}^{(g_1)} s_{k_2}^{(g_2)} \cdots s_{k_j}^{(g_j)} \,, \qquad s_k^{(g)} = \begin{cases} \tilde{s}_k \,, & g = 1 \,, \\ -s_k \,, & g = 2 \,, \end{cases}
$$

where $\tilde{s}_k = -\frac{ab}{c'}Q^{k-\frac{1}{2}} - \frac{a'c'}{b}Q^{\frac{1}{2}-k}$ and $s_k = (1-Q^{-k})(cc'-bb'Q^k)$. The coefficient $Z_n$ can be interpreted as the $n$-body partition function for particles in a one-body spectrum $\epsilon_k$ ($k = 1, \ldots, q$) obeying a mixture of two statistics: Fermions with Boltzmann factor $e^{-\beta \epsilon_k} = \tilde{s}_k$, and two-fermion bound states occupying one-body energy levels $k$ and $k+1$ with Boltzmann factor $e^{-\beta \epsilon_{k,k+1}} = -s_k$ behaving effectively as $g = 2$ exclusons. Following the same reasoning as in Section 2.2, the associated cluster coefficient $b_n$, defined by $\log\left(\sum_{n=0}^{q} Z_n z^n\right) = \sum_{n=1}^{\infty} b_n z^n$, can

be expressed in terms of $\tilde{s}_k$ and $s_k$ and the full generating function for signed area enumeration of triangular lattice walks can be derived from the cluster coefficient, that is, for $N < q$,

$$\text{Tr}\, H_{\text{tri}}^N = \frac{1}{q}\text{tr}\, H_{1,2}^N = N(-1)^{N+1}\frac{1}{q}b_N$$

$$= N \sum_{\substack{\tilde{l}_1,\ldots,\tilde{l}_{j+1};l_1,\ldots,l_j \\ (1,2)\text{-composition of } N}} c_{1,2}(\tilde{l}_1,\ldots,\tilde{l}_{j+1};l_1,\ldots,l_j)\frac{1}{q}\sum_{k=1}^{q-j}\tilde{s}_k^{\tilde{l}_1}s_k^{l_1}\tilde{s}_{k+1}^{\tilde{l}_2}s_{k+1}^{l_2}\cdots\tilde{s}_{k+j}^{\tilde{l}_{j+1}}, \qquad (7)$$

where the combinatorial coefficients

$$c_{1,2}(\tilde{l}_1,\ldots,\tilde{l}_{j+1};l_1,\ldots,l_j) = \frac{(\tilde{l}_1+l_1-1)!}{\tilde{l}_1!\, l_1!}\prod_{k=2}^{j+1}\binom{l_{k-1}+\tilde{l}_k+l_k-1}{l_{k-1}-1,\ \tilde{l}_k,\ l_k}$$

process a combinatorial interpretation of periodic Motzkin path counting (See [46]). By convention, $l_k = 0$ for $k > j$. The sequence of integers $\tilde{l}_1,\ldots,\tilde{l}_{j+1};l_1,\ldots,l_j$, $j \geq 0$, labeling $c_{1,2}$ in (7) is defined as a $(1,2)$-composition of the integer $N$ if they satisfy the conditions $N = (\tilde{l}_1+\tilde{l}_2+\cdots+\tilde{l}_{j+1})+2(l_1+l_2+\cdots+l_j)$ with $\tilde{l}_i \geq 0$ and $l_i > 0$. That is, the $l_i$'s are the usual compositions of integers $1,2,\ldots,\lfloor N/2\rfloor$, while the $\tilde{l}_i$'s are additional nonnegative integers (for $j = 0$, we have the trivial composition $\tilde{l}_1 = N$). For example, there are six compositions of 4: (4), (2,0;1), (1,1;1), (0,2;1), (0,0;2), (0,0,0;1,1).

For the trigonometric sum $\frac{1}{q}\sum_{k=1}^{q-j}\tilde{s}_k^{\tilde{l}_1}s_k^{l_1}\tilde{s}_{k+1}^{\tilde{l}_2}s_{k+1}^{l_2}\cdots\tilde{s}_{k+j}^{\tilde{l}_{j+1}}$, we consider a special case where $a' = ab^2/c'^2$, $b' = cc'/b$, which includes the specific choices $(a,a',b,b',c,c') = (a,a,b,b,b,b)$ and $(1,a',1,a'^{-1/2},1,a'^{-1/2})$. Let $ab/c' = a'c'/b = \lambda_1$ and $bb' = cc' = \lambda_2$. The two spectral functions are expressed as

$$\tilde{s}_k = \lambda_1(S_{k-\frac{1}{2}}-2), \qquad s_k = \lambda_2 S_k,$$

with $S_k = 4\sin^2(k\pi p/q)$ the standard Hofstadter spectral function. By expanding the trigonometric sum using the binomial theorem, we obtain

$$\text{Tr}\, H_{\text{tri}}^N = N \sum_{\substack{\tilde{l}_1,\ldots,\tilde{l}_{j+1};l_1,\ldots,l_j \\ (1,2)\text{-composition of } n=0,1,2,\ldots,N}} c_{1,2}(\tilde{l}_1,\ldots,\tilde{l}_{j+1};l_1,\ldots,l_j)(-2)^{N-n}\lambda_1^{N-2(l_1+\cdots+l_j)}\lambda_2^{l_1+\cdots+l_j}$$

$$\times \binom{N-1}{n-1}\sum_{k=1}^{q-j}S_{k-\frac{1}{2}}^{\tilde{l}_1}S_k^{l_1}S_{k+\frac{1}{2}}^{\tilde{l}_2}S_{k+1}^{l_2}\cdots S_{k+j-\frac{1}{2}}^{\tilde{l}_{j+1}}.$$

By convention, for $(1,2)$-composition of $n = 0$, $c_{1,2}(0)\binom{N-1}{n-1}$ is understood to be $\frac{1}{N}$. Thanks to the identity

$$\sum_{k=1}^{q-j}S_{2k-1}^{\tilde{l}_1}S_{2k}^{l_1}S_{2k+1}^{\tilde{l}_2}S_{2k+2}^{l_2}\cdots S_{2k+2j-1}^{\tilde{l}_{j+1}} = \sum_{k=1}^{q-2j-1}S_k^{\tilde{l}_1}S_{k+1}^{l_1}\cdots S_{k+2j}^{\tilde{l}_{j+1}},$$

the generating function $C_N(A)$ for the enumeration of closed triangular lattice walks of length $N$ enclosing a signed area[3] $A$ ($A = 1/2$ for a unit lattice cell), and characterized by the number

---

[3]The area $2A = 0, \pm 1, \pm 2, \ldots$ is bounded by round$(N^2/6)-(0$ if $N \equiv 0 \pmod 6$, 1 otherwise) [48]. The function round$(x)$ returns the integer that is closest to $x$ and rounds half-integers towards the nearest even integer.

of moves in the six directions, is derived to be

$$
C_N(A) = N \sum_{\substack{\tilde{l}_1,\dots,\tilde{l}_{j+1};l_1,\dots,l_j \\ (1,2)\text{-composition of } n=0,1,2,\dots,N}} \frac{(\tilde{l}_1+l_1-1)!}{\tilde{l}_1!\, l_1!} \prod_{k=2}^{j+1} \binom{l_{k-1}+\tilde{l}_k+l_k-1}{l_{k-1}-1,\ \tilde{l}_k,\ l_k} (-2)^{N-n} \lambda_1^{N-2(l_1+\cdots+l_j)} \lambda_2^{l_1+\cdots+l_j}
$$

$$
\times \binom{N-1}{n-1} \sum_{k_3=-\tilde{l}_2}^{\tilde{l}_2} \sum_{k_4=-l_2}^{l_2} \cdots \sum_{k_{2j+1}=-\tilde{l}_{j+1}}^{\tilde{l}_{j+1}} \binom{2\tilde{l}_1}{\tilde{l}_1+2A+\sum_{i=3}^{2j+1}(i-2)k_i} \binom{2l_1}{l_1-2A-\sum_{i=3}^{2j+1}(i-1)k_i}
$$

$$
\times \prod_{i=2}^{j+1} \binom{2\tilde{l}_i}{\tilde{l}_i+k_{2i-1}} \prod_{i=2}^{j} \binom{2l_i}{l_i+k_{2i}}. \tag{8}
$$

Similar to the square lattice case, (8) retains a three-part structure: A summation indexed by the $(1,2)$-composition, an associated combinatorial factor, and a nested multiple sum derived from two spectral functions. For $a = a' = 0$ or $\lambda_1 = 0$, the terms in $C_N(A)$ are nonzero only if $N - 2(l_1 + \cdots + l_j) = 0$. This implies that all $\tilde{l}_i$'s are zero, and we recover the $g = 2$ square lattice case.

The general case with arbitrary variables $a, a', b, b', c, c'$ can be treated in a similar approach. However, the resulting expression is cumbersome and will not be presented in this paper. A few examples of $\mathbf{Tr}\, H_{\text{tri}}^N$ are listed below, with the corresponding $C_N(A)$ values for $a = a' = b = b' = c = c' = 1$ listed in Table 2 in Appendix B.

$$
\mathbf{Tr}\, H_{\text{tri}}^2 = 2(aa' + bb' + cc'),
$$

$$
\mathbf{Tr}\, H_{\text{tri}}^3 = 3(Q^{1/2} + Q^{-1/2})(abc + a'b'c'),
$$

$$
\mathbf{Tr}\, H_{\text{tri}}^4 = 6\big[(aa')^2 + (bb')^2 + (cc')^2\big] + \big[16 + 4(Q + Q^{-1})\big](aa'bb' + bb'cc' + cc'aa'),
$$

$$
\mathbf{Tr}\, H_{\text{tri}}^5 = \big[25(Q^{1/2} + Q^{-1/2}) + 5(Q^{3/2} + Q^{-3/2})\big](abc + a'b'c')(aa' + bb' + cc'),
$$

$$
\begin{aligned}
\mathbf{Tr}\, H_{\text{tri}}^6 = {}& 20\big[(aa')^3 + (bb')^3 + (cc')^3\big] \\
& + \big[36 + 21(Q + Q^{-1}) + 6(Q^2 + Q^{-2})\big]\big[(abc)^2 + (a'b'c')^2\big] \\
& + \big[96 + 36(Q + Q^{-1}) + 6(Q^2 + Q^{-2})\big] \\
& \quad \times \big[(aa')^2(bb' + cc') + (bb')^2(cc' + aa') + (cc')^2(aa' + bb')\big] \\
& + \big[312 + 162(Q + Q^{-1}) + 36(Q^2 + Q^{-2}) + 6(Q^3 + Q^{-3})\big]aa'bb'cc',
\end{aligned}
$$

$$
\begin{aligned}
\mathbf{Tr}\, H_{\text{tri}}^7 = {}& 7\big\{\big[22(Q^{1/2} + Q^{-1/2}) + 7(Q^{3/2} + Q^{-3/2}) + (Q^{5/2} + Q^{-5/2})\big]\big[(aa')^2 + (bb')^2 + (cc')^2\big] \\
& + \big[60(Q^{1/2} + Q^{-1/2}) + 24(Q^{3/2} + Q^{-3/2}) + 5(Q^{5/2} + Q^{-5/2}) + (Q^{7/2} + Q^{-7/2})\big] \\
& \quad \times (aa'bb' + bb'cc' + cc'aa')\big\}(abc + a'b'c').
\end{aligned}
$$

When $a = a' = 0$ (or equivalently $b = b' = 0$ or $c = c' = 0$), the enumeration reduces to the case of square lattice walks. When $a' = b' = c' = 0$ (or equivalently $a = b = c = 0$), it simplifies to the enumeration of *chiral* walks on a triangular lattice, which has been proven to be related to $g = 3$ exclusion statistics for the isotropic case $a = b = c = 1$ [27, 29]. It can be further shown that for the anisotropic case $g = 3$ exclusion statistics still hold, but now with the spectral function $s_k = 4abc \sin(\pi kp/q) \sin(\pi(k+1)p/q)$.

## 3 Quantum A-period conjecture

Quantum periods, expressed as formal power series in $\hbar^2$, emerge from the all-orders WKB method and encapsulate quantum corrections to classical periods on complex curves. They play a pivotal role in the study of quantum curves, appearing in various settings such as

Schrödinger operators and mirror curves of toric Calabi–Yau geometries. Their significance lies in their role in exact quantization conditions, mirror symmetry, resurgence theory, and connections to BPS state counting in supersymmetric and string theory contexts. See, e.g., [49,50] for comprehensive reviews. Specifically, the quantum A-period, in the limit $E \to \infty$ (i.e., around $z = 1/E = 0$), can be obtained by a residue calculation [34,51,52].

Following the procedure in [34,53], we promote canonically conjugate variables $x$ and $y$ in the mirror curve of local $\mathcal{B}_3$ geometry

$$\mathrm{e}^x + \mathrm{e}^y + \mathrm{e}^{-x-y} + m_1 \mathrm{e}^{-x} + m_2 \mathrm{e}^{-y} + m_3 \mathrm{e}^{x+y} = E,$$

where $m_1$, $m_2$, and $m_3$ are complex moduli parameters, to two operators $\mathsf{x}$ and $\mathsf{y}$ with the commutation relation $[\mathsf{x}, \mathsf{y}] = \mathrm{i}\hbar$. This leads to the Hamiltonian

$$H_{\mathcal{B}_3} = \mathrm{e}^{\mathsf{x}} + \mathrm{e}^{\mathsf{y}} + \mathrm{e}^{-\mathsf{x}-\mathsf{y}} + m_1 \mathrm{e}^{-\mathsf{x}} + m_2 \mathrm{e}^{-\mathsf{y}} + m_3 \mathrm{e}^{\mathsf{x}+\mathsf{y}},$$

which can be seen as a non-compact version of $H_{\mathrm{tri}}$ in (2). Applying the Baker–Campbell–Hausdorff formula, the Schrödinger equation $H_{\mathcal{B}_3}\psi(x) = E\psi(x)$ leads the quantized mirror curve

$$\mathrm{e}^x\psi(x) + \psi(x-\mathrm{i}\hbar) + Q^{-1/2}\mathrm{e}^{-x}\psi(x+\mathrm{i}\hbar) + m_1\mathrm{e}^{-x}\psi(x) + m_2\psi(x+\mathrm{i}\hbar) + m_3 Q^{-1/2}\mathrm{e}^x\psi(x-\mathrm{i}\hbar) = E\psi(x), \quad (9)$$

where $Q = \mathrm{e}^{\mathrm{i}\hbar}$. Introducing $V(X) = \frac{\psi(x-\mathrm{i}\hbar)}{\psi(x)}$ with $X = \mathrm{e}^x$, (9) becomes

$$X + V(X) + \frac{Q^{-1/2}}{XV(QX)} + \frac{m_1}{X} + \frac{m_2}{V(QX)} + m_3 Q^{-1/2} X V(X) = \frac{1}{z}, \qquad z = \frac{1}{E}.$$

Express $V(X) = v_{-1}(X)/z + v_0(X) + v_1(X)z + \cdots$. The quantum A-period is given by [34]

$$
\begin{aligned}
t &= -\log(z) + \operatorname*{Res}_{X=0} \frac{\log V(X) - \log(v_{-1}(X)/z)}{X} \\
&= -\log(z) - z^2(m_1 + m_2 + m_3) - z^3(Q^{1/2} + Q^{-1/2})(1 + m_1 m_2 m_3) \\
&\quad - z^4\left[\frac{3}{2}\left(m_1^2 + m_2^2 + m_3^2\right) + (4 + Q + Q^{-1})(m_1 m_2 + m_2 m_3 + m_3 m_1)\right] + \mathcal{O}(z^5). \quad (10)
\end{aligned}
$$

We observe that (10) is related to the signed area enumeration of triangular lattice walks, namely

$$t = -\log(z) - \sum_{N=1}^{\infty} z^N \frac{1}{N} \mathbf{Tr}\, H_{\mathrm{tri}}^N = -\log(z) - \sum_{N=1}^{\infty} z^N \frac{1}{N} \sum_A C_N(A) Q^A, \quad (11)$$

where $C_N(A)$ is the generating function for the enumeration of closed random walks on a triangular lattice with $N$ steps, enclosing a signed area $A$ ($A = 1/2$ for a unit lattice cell), and characterized by the variables $(a, a', b, b', c, c') = (1, m_1, 1, m_2, 1, m_3)$ for counting the moves. The conjecture (11) has been verified up to $z^{18}$. For $Q \to 1$, the quantum A-period reduces to the classical A-period $-\log(z) - \sum_{N=1}^{\infty} z^N \frac{1}{N}[x^0 y^0](x + m_1 x^{-1} + y + m_2 y^{-1} + x^{-1}y^{-1} + m_3 xy)^N$. From (7), we obtain

$$t = (1/q)\log\det(1/z - H_{1,2}) + \mathcal{O}(z^q),$$

where $H_{1,2}$ is the $q \times q$ tridiagonal matrix representing the effective Hofstadter Hamiltonian for triangular lattice walks, as defined in Section 2.3.

The conjecture (11) establishes a compelling connection between quantities in toric Calabi–Yau threefolds and lattice random walks. To the best of the author's knowledge, the statistical mechanics methods arising from Hofstadter-like models have not yet been applied to the study of the toric Calabi–Yau geometries. This novel perspective has the potential to bridge

toric string theory, condensed matter physics, and statistical mechanics, fostering deeper cross-disciplinary understanding and illuminating potential applications across these fields.

We also observe that (11) holds for other toric Calabi–Yau threefolds, such as local $\mathbb{F}_0$ geometry, which has been shown to be related to the Hofstadter model [33] and thus square lattice walks, where $H_{\text{tri}}$ becomes $H_{\text{sq}}$ and $C_N(A)$ represents the generating function for enumeration of closed random walks on a square lattice with $N$ steps and enclosing a signed area $A$ ($A = 1$ for a unit lattice cell), characterized by the number of moves in each direction. We can check that the quantum A-period derived from the Hamiltonian associated with local $\mathbb{F}_0$

$$H_{\mathbb{F}_0} = e^x + e^{-x} + R^2(e^y + e^{-y}),\tag{12}$$

where $R$ is a parameter, reads

$$t = -\log(z) - z^2(1 + R^4) - z^4\left[\frac{3}{2} + 4R^4 + \frac{3}{2}R^8 + R^4(Q + Q^{-1})\right] - z^6\left[\frac{10}{3} + 16R^4\right.$$
$$\left. + 16R^8 + \frac{10}{3}R^{12} + 6R^4(1 + R^4)(Q + Q^{-1}) + R^4(1 + R^4)(Q^2 + Q^{-2})\right] + \mathcal{O}(z^8),$$

which is consistent with (11) for $(b, b', c, c') = (1, 1, R^2, R^2)$ in $H_2$.

In the square lattice case, we can go a step further by deriving the right-hand side of (11). By applying the method in [47, 54] and using the identity $[x^0y^0](bx + b'x^{-1} + cy + c'y^{-1})^{2n} = \sum_{k=0}^n \binom{2n}{n}\binom{n}{k}^2(bb')^{n-k}(cc')^k$, where $x$ and $y$ are scalars, we find

$$G(z) = \sum_{N=0}^{\infty} z^N \sum_A C_N(A)Q^A = -\frac{z\,\partial_z\det(1/z - H_2)}{2\pi q(bb'cc')^{q/4}}\frac{K(1/f(z))}{\sqrt{f(z)}},$$

where

$$f(z) = \frac{(\det(1/z - H_2))^2 - 4[(bb')^{q/2} - (cc')^{q/2}]^2}{16(bb'cc')^{q/2}},$$

and $K$ denotes the complete elliptic integral of the first kind. By convention, there is one walk with signed area 0 for $N = 0$. From (11), the quantum A-period is given by

$$t = -\log(z) - \int_0^z \frac{G(u) - 1}{u}du,$$

and its derivative, by replacing $z = 1/E$, reads

$$\frac{\partial t}{\partial E} = \frac{1}{E}G\left(\frac{1}{E}\right) = \frac{\partial_E\det(E - H_2)}{2\pi q(bb'cc')^{q/4}}\frac{K(1/F)}{\sqrt{F}},$$

with

$$F = f\left(\frac{1}{E}\right) = \frac{(\det(E - H_2))^2 - 4[(bb')^{q/2} - (cc')^{q/2}]^2}{16(bb'cc')^{q/2}}.$$

For $(b, b', c, c') = (1, 1, R^2, R^2)$, we recover Eq. (3.22) in [33], up to a factor 2 arising from differing definitions of the classical period.[4] Note that the polynomial $P_{p/q}(E, R)$ in Eq. (2.21)

---

[4]We thank Yasuyuki Hatsuda for clarifying this point.

of [33] is nothing but $\det(E - H_2)$, leading to the strong-weak coupling energy relation[5]

$$\det(E_n - H_2) = \det(\tilde{E}_n - \tilde{H}_2), \tag{13}$$

where $E_n$ and $\tilde{E}_n$ are the $n$-th eigenenergy of the Hamiltonian $H_{\mathbb{F}_0}$ in (12) and its dual $e^{2\pi x/\hbar} + e^{-2\pi x/\hbar} + R^{4\pi/\hbar}(e^{2\pi y/\hbar} + e^{-2\pi y/\hbar})$ with $[x, y] = i\hbar$ and $\hbar = 2\pi p/q$. The Hamiltonians $H_2$, $\tilde{H}_2$ are the $q \times q$ and $p \times p$ matrices, as defined in Section 2.2, with $Q = e^{2\pi i p/q}$ and $e^{2\pi i q/p}$, $(b, b', c, c') = (1, 1, R^2, R^2)$ and $(1, 1, R^{2q/p}, R^{2q/p})$, respectively. In Section 2.2, the secular determinant $\det(I - zH_2)$ is expanded in terms of the Kreft coefficients, which can be interpreted as the grand partition function of a system of $g = 2$ exclusons. Consequently, the same coefficients emerge in the expansion of $\det(E_n - H_2) = E_n^q \det(I - E_n^{-1} H_2)$ in (13). These coefficients have recently been recovered and proved in [55] for the case $R = 1$ using quantum group theory. However, the role of the strong-weak energy relation (13) within the framework of statistical mechanics remains unclear. Uncovering its connection to statistical mechanics could provide valuable new insights into understanding the symmetric property of the quantization condition for local $\mathbb{F}_0$ [33, 56].

## 4  Conclusion

In this paper, by calculating the trace of powers of anisotropic Hofstadter-like Hamiltonians, we obtain closed-form expressions (6) and (8) for the enumeration of square and triangular lattice walks with a given length and signed area. These results generalize the previous counting formulae for the isotropic case in [26–28] by also accounting for the number of moves in each hopping direction. Similar to the isotropic case, both square and triangular lattice walks relate to exclusion statistics: The former corresponds to $g = 2$, while the latter is a mixture of $g = 1$ and $g = 2$. Note that the enumeration can also be computed recursively, as detailed in the Appendix C. Nevertheless, our method offers a perspective that connects lattice random walks with exclusion statistics, providing novel mathematical insights into Hofstadter-like spectral structures and serving as a useful analytical tool for the study of complex systems in statistical and polymer physics.

The relation (11) between $C_N(A)$ and the quantum A-period in toric Calabi–Yau threefolds is observed. While this connection is not entirely unexpected, as both originate from the same quantum curve, the simple and concise form of this relation suggests an underlying structure. We have verified several implications of this relation using established results in the literature. Similar verification could be pursued for closed random walks on other planar lattices (such as the honeycomb lattice, Lieb lattice, king's lattice, and kagome lattice) and their associated toric Calabi–Yau threefolds. The proof of this conjecture, along with its broader consequences, stands as an open question inviting deeper exploration. One natural approach would be to verify the conjecture (11) by demonstrating that both sides satisfy the same difference equation. It is well known that the classical period satisfies the Picard–Fuchs differential equation. However, to our knowledge, the quantization of the Picard–Fuchs equation, which would lead to

---

[5]To numerically calculate the eigenenergies $E_n$ and $\tilde{E}_n$, we can choose $x = \sqrt{\hbar/2}\,(a^\dagger + a)$ and $y = i\sqrt{\hbar/2}\,(a^\dagger - a)$, where $a^\dagger$ and $a$ are creation and annihilation operators, represented by the matrices

$$a^\dagger = \begin{pmatrix} 0 & 0 & 0 & 0 & \cdots \\ 1 & 0 & 0 & 0 & \cdots \\ 0 & \sqrt{2} & 0 & 0 & \cdots \\ 0 & 0 & \sqrt{3} & 0 & \cdots \\ \vdots & \vdots & \vdots & \vdots & \ddots \end{pmatrix}, \qquad a = \begin{pmatrix} 0 & 1 & 0 & 0 & \cdots \\ 0 & 0 & \sqrt{2} & 0 & \cdots \\ 0 & 0 & 0 & \sqrt{3} & \cdots \\ 0 & 0 & 0 & 0 & \cdots \\ \vdots & \vdots & \vdots & \vdots & \ddots \end{pmatrix},$$

respectively, which are of large, truncated finite dimension, ensuring sufficient accuracy.

a corresponding difference equation, has not yet been explored. This quantum counterpart of the classical Picard–Fuchs equation is also expected to govern the generating function for the signed area enumeration, which acts as a recurrence relation for $C_N(A)$ up to a factor.

Furthermore, the concept of signed area has recently been extended to three dimensional walks, such as closed cubic lattice walks [57], where the signed area is defined as the sum of signed areas obtained from the walk's projection onto the three Cartesian planes. The enumeration formula can also be mapped onto cluster coefficients, now involving three types of particles. Investigating its potential connection to a toric Calabi–Yau *fourfold* represents another promising direction for future work. Finally, the potential relation between lattice random walks and the quantum B-period poses an intriguing challenge for further investigation.

## Acknowledgments

L.G. would like to thank Stéphane Ouvry, Alexios P. Polychronakos, Hai-Long Shi, and Masahito Yamazaki for insightful discussions, as well as the anonymous referees for their useful comments and suggestions. L.G. also acknowledges the hospitality of the Laboratory of Theoretical Physics and Statistical Models (LPTMS), a CNRS–Université Paris-Saclay joint research unit, during the early stages of this work.

**Funding information** L.G. was supported by the GGI Boost Postdoctoral Fellowship (No. 25768/2023).

## A   Examples of signed area enumeration of square lattice walks

Table 1: $C_N(A)$ up to $N = 18$ for closed $N$-step square lattice walks with signed area $A$ ($A = 1$ for a unit lattice cell) and $b = b' = c = c' = 1$.

| | $N = 2$ | 4 | 6 | 8 | 10 | 12 | 14 | 16 | 18 |
|---|---|---|---|---|---|---|---|---|---|
| $A = 0$ | 4 | 28 | 232 | 2156 | 21944 | 240280 | 2787320 | 33820044 | 424925872 |
| ±1 | | 8 | 144 | 2016 | 26320 | 337560 | 4337088 | 56267456 | 739225296 |
| ±2 | | | 24 | 616 | 11080 | 174384 | 2582440 | 37139616 | 526924440 |
| ±3 | | | | 96 | 3120 | 67256 | 1220464 | 20255488 | 319524480 |
| ±4 | | | | 16 | 840 | 23928 | 525224 | 10030216 | 176290488 |
| ±5 | | | | | 160 | 7272 | 203952 | 4579520 | 90612576 |
| ±6 | | | | | 40 | 2400 | 80752 | 2072736 | 45522456 |
| ±7 | | | | | | 528 | 27440 | 870080 | 21840912 |
| ±8 | | | | | | 144 | 9800 | 368208 | 10416744 |
| ±9 | | | | | | 24 | 3024 | 146112 | 4797504 |
| ±10 | | | | | | | 840 | 56128 | 2171448 |
| ±11 | | | | | | | 224 | 20672 | 956016 |
| ±12 | | | | | | | 56 | 7520 | 417456 |
| ±13 | | | | | | | | 2176 | 168624 |
| ±14 | | | | | | | | 704 | 69120 |
| ±15 | | | | | | | | 192 | 26784 |
| ±16 | | | | | | | | 32 | 9576 |
| ±17 | | | | | | | | | 3168 |
| ±18 | | | | | | | | | 1080 |
| ±19 | | | | | | | | | 288 |
| ±20 | | | | | | | | | 72 |
| Total | 4 | 36 | 400 | 4900 | 63504 | 853776 | 11778624 | 165636900 | 2363904400 |

We note that, since

$$N \sum_{\substack{l_1, l_2, \ldots, l_j \\ \text{composition of } N/2}} c_2(l_1, l_2, \ldots, l_j) = \binom{N}{N/2},$$

and in the limit $q \to \infty$, i.e., $Q \to 1$ [26, 29],

$$\frac{1}{q} \sum_{k=1}^{q-j} S_k^{l_1} S_{k+1}^{l_2} \cdots S_{k+j-1}^{l_j} \to \binom{2(l_1 + l_2 + \cdots + l_j)}{l_1 + l_2 + \cdots + l_j}, \tag{A.1}$$

the total number of closed square lattice walks of length $N$ is recovered to be

$$N \sum_{\substack{l_1, l_2, \ldots, l_j \\ \text{composition of } N/2}} c_2(l_1, l_2, \ldots, l_j) \binom{2(l_1 + l_2 + \cdots + l_j)}{l_1 + l_2 + \cdots + l_j} = \binom{N}{N/2}^2,$$

as expected.[6]

## B  Examples of signed area enumeration of triangular lattice walks

Table 2: $C_N(A)$ up to $N = 12$ for closed $N$-step triangular lattice walks with signed area $A$ ($A = 1/2$ for a unit lattice cell) and $a = a' = b = b' = c = c' = 1$.

| | $N=2$ | 3 | 4 | 5 | 6 | 7 | 8 | 9 | 10 | 11 | 12 |
|---|---|---|---|---|---|---|---|---|---|---|---|
| $2A=$ 0 | 6 | | 66 | | 1020 | | 19890 | | 449976 | | 11177244 |
| $\pm 1$ | | 12 | | 300 | | 6888 | | 164124 | | 4124340 | |
| $\pm 2$ | | | 24 | | 840 | | 23904 | | 654840 | | 18038232 |
| $\pm 3$ | | | | 60 | | 2604 | | 85944 | | 2617428 | |
| $\pm 4$ | | | | | 168 | | 8568 | | 317940 | | 10572216 |
| $\pm 5$ | | | | | | 504 | | 29628 | | 1215456 | |
| $\pm 6$ | | | | | 12 | | 1968 | | 114360 | | 4919592 |
| $\pm 7$ | | | | | | 84 | | 8496 | | 475200 | |
| $\pm 8$ | | | | | | | 432 | | 37560 | | 2058096 |
| $\pm 9$ | | | | | | | | 1980 | | 167244 | |
| $\pm 10$ | | | | | | | 48 | | 10380 | | 785976 |
| $\pm 11$ | | | | | | | | 432 | | 55308 | |
| $\pm 12$ | | | | | | | | | 2700 | | 288276 |
| $\pm 13$ | | | | | | | | 36 | | 15972 | |
| $\pm 14$ | | | | | | | | | 540 | | 96840 |
| $\pm 15$ | | | | | | | | | | 4356 | |
| $\pm 16$ | | | | | | | | | 60 | | 30312 |
| $\pm 17$ | | | | | | | | | | 924 | |
| $\pm 18$ | | | | | | | | | | | 8544 |
| $\pm 19$ | | | | | | | | | | 132 | |
| $\pm 20$ | | | | | | | | | | | 2088 |
| $\pm 22$ | | | | | | | | | | | 336 |
| $\pm 24$ | | | | | | | | | | | 24 |
| Total | 6 | 12 | 90 | 360 | 2040 | 10080 | 54810 | 290640 | 1588356 | 8676360 | 47977776 |

---

[6]By setting $Q = 1$, the total number of $N$-step closed square lattice walks is obtained from the $u, v$-independent part in the expansion of $(u + u^{-1} + v + v^{-1})^N$. Since $vu = uv$, the binomial theorem gives

$$(u + u^{-1} + v + v^{-1})^N = (u + v)^N (1 + u^{-1}v^{-1})^N = \sum_{i=0}^{N} \sum_{j=0}^{N} \binom{N}{i}\binom{N}{j} u^i v^{N-i} (vu)^{-j}.$$

Taking $i = j = N/2$, the count simplifies to $\binom{N}{N/2}^2$.

We note that, from

$$
n \sum_{\substack{\tilde{l}_1,\ldots,\tilde{l}_{j+1};l_1,\ldots,l_j \\ (1,2)\text{-composition of } n \\ l_1+\cdots+l_j=n'}} c_{1,2}(\tilde{l}_1,\ldots,\tilde{l}_{j+1};l_1,\ldots,l_j) = \binom{n}{2n'}\binom{2n'}{n'},
$$

the total number of closed triangular lattice walks of length $N$ is recovered to be

$$
N \sum_{\substack{\tilde{l}_1,\ldots,\tilde{l}_{j+1};l_1,\ldots,l_j \\ (1,2)\text{-composition of } n=0,1,2,\ldots,N}} c_{1,2}(\tilde{l}_1,\ldots,\tilde{l}_{j+1};l_1,\ldots,l_j)(-2)^{N-n}\binom{N-1}{n-1}\binom{2(\tilde{l}_1+\cdots+\tilde{l}_{j+1}+l_1+\cdots+l_j)}{\tilde{l}_1+\cdots+\tilde{l}_{j+1}+l_1+\cdots+l_j}
$$

$$
= \sum_{n=0}^{N}\sum_{n'=0}^{\lfloor n/2\rfloor}\left(N \sum_{\substack{\tilde{l}_1,\ldots,\tilde{l}_{j+1};l_1,\ldots,l_j \\ (1,2)\text{-composition of } n \\ l_1+\cdots+l_j=n'}} c_{1,2}(\tilde{l}_1,\ldots,\tilde{l}_{j+1};l_1,\ldots,l_j)(-2)^{N-n}\binom{N-1}{n-1}\binom{2n-2n'}{n-n'}\right)
$$

$$
= \sum_{n=0}^{N}\sum_{n'=0}^{\lfloor n/2\rfloor}(-2)^{N-n}\binom{N}{n}\binom{n}{2n'}\binom{2n'}{n'}\binom{2n-2n'}{n-n'}
$$

$$
= \sum_{n=0}^{N}\sum_{k=0}^{n}(-2)^{N-n}\binom{N}{n}\binom{n}{k}^3,
$$

as expected.[7]

## C  Recurrence relation for enumeration of triangular lattice walks

Consider an $N$-step random walk (not necessarily closed) on the deformed triangular lattice (Figure 2, right) with $m_1$ steps right, $m_2$ steps left, $l_1$ steps up, $l_2$ steps down, $r_1$ steps down-left, and $r_2$ steps up-right with $m_1+m_2+l_1+l_2+r_1+r_2=N$. If the walk is open, we can close it by adding a straight line that connects the endpoint to the starting point. By convention, the area of a unit lattice cell is $1/2$. Let $C_{m_1,m_2,l_1,l_2,r_1,r_2}(A)$ denote the generating function that counts $N$-step walks with signed area $A$, characterized by the number of moves in each possible direction. The full generating function $Z_{m_1,m_2,l_1,l_2,r_1,r_2}(Q) = \sum_A C_{m_1,m_2,l_1,l_2,r_1,r_2}(A)Q^A$ can be computed by the recurrence relation

$$
Z_{m_1,m_2,l_1,l_2,r_1,r_2}(Q)
$$
$$
= a\,Q^{(l_2+r_1-l_1-r_2)/2}Z_{m_1-1,m_2,l_1,l_2,r_1,r_2}(Q) + a'\,Q^{(l_1+r_2-l_2-r_1)/2}Z_{m_1,m_2-1,l_1,l_2,r_1,r_2}(Q)
$$
$$
+ b\,Q^{(m_1+r_2-m_2-r_1)/2}Z_{m_1,m_2,l_1-1,l_2,r_1,r_2}(Q) + b'\,Q^{(m_2+r_1-m_1-r_2)/2}Z_{m_1,m_2,l_1,l_2-1,r_1,r_2}(Q)
$$
$$
+ c\,Q^{(m_2-m_1+l_1-l_2)/2}Z_{m_1,m_2,l_1,l_2,r_1-1,r_2}(Q) + c'\,Q^{(m_1-m_2+l_2-l_1)/2}Z_{m_1,m_2,l_1,l_2,r_1,r_2-1}(Q),
$$

---

[7]By setting $Q = 1$, the total number of $N$-step closed triangular lattice walks can also be derived from the $u,v$-independent part in the expansion of

$$
(u+u^{-1}+v+v^{-1}+u^{-1}v^{-1}+vu)^N = [(1+u)(1+v)(1+u^{-1}v^{-1})-2]^N
$$
$$
= \sum_{n=0}^{N}(-2)^{N-n}\binom{N}{n}\sum_{i=0}^{n}\sum_{j=0}^{n}\sum_{k=0}^{n}\binom{n}{i}\binom{n}{j}\binom{n}{k}u^i v^j (vu)^{-k}.
$$

Setting $i = j = k$ yields the desired number.

with $Z_{0,0,0,0,0,0}(Q) = 1$ and $Z_{m_1,m_2,l_1,l_2,r_1,r_2}(Q) = 0$ whenever $\min(m_1, m_2, l_1, l_2, r_1, r_2) < 0$. For closed walks of length $N$, we have

$$\sum_A C_N(A) Q^A = \sum_{\substack{m_1+m_2+l_1+l_2+r_1+r_2=N \\ m_1+r_2=m_2+r_1 \\ l_1+r_2=l_2+r_1}} Z_{m_1,m_2,l_1,l_2,r_1,r_2}(Q).$$

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
