# Peer review of "Lattice random walks and quantum A-period conjecture"

_SciPost Physics, doi:SciPost Phys. 19, 053 (2025)_

## Round 1 · Referee Report · Anonymous (Referee 1) · 2025-3-24

Report

This papers derives expressions for the number of closed walks on a triangular lattice with given area and number of steps in each direction, and presents a conjecture on the relation of the quantum A-period of Calabi-Yau threefolds and lattice walks. The paper is interesting and it should be published. I have, nevertheless, a few remarks.

  1. The parameters a, a',b,b',c,c' introduced to enumerate closed paths of various numbers of steps in each direction seem redundant: since only closed walks are considered, the number of steps in each direction are restricted. This is born out by the fact that in all formulas for the square lattice only the products aa' and bb' appear, and in all formulas for the triangular lattice only aa', bb', cc' and abc appear. It would be simpler to use the minimal number of parameters by taking, say, a=1 and b=1.

  2. In p.6, to eliminate the corner elements of the matrix it is stated that "we set (...) and ignore g_q". I do not understand what "ignoring g_q" means: it either vanishes or it does not, and here it does not, so the second term in the parenthesis of the "spurious" term does not vanish. Also, I believe the relation (...) for the vanishing of f_q is missing a minus sign. Similar remarks apply to the beginning of section 2.3.

  3. The conjecture of section 3, eq. (11), is definitely interesting. Given the close connection of the Hofstadter-like Hamiltonian and the one for the mirror curve, it should be possible to actually prove the conjecture, and I am a bit surprised that the author did not push the analysis to provide such a proof. A comment on this in the paper would be useful.

The paper can be published after the author had a chance to consider the above points and make any related modifications.

Recommendation

Ask for minor revision

  • validity: -
  • significance: -
  • originality: -
  • clarity: -
  • formatting: -
  • grammar: -

Author:  Li Gan  on 2025-05-21  [id 5503]

(in reply to Report 1 on 2025-03-24)

We thank the Referee for the helpful comments and suggestions.

  1. We agree that the introduced parameters are redundant and can be reduced, for example, by setting $a=b=1$ for both closed square and triangular lattice walks. However, we prefer to retain them to make the step counts in each direction explicit. This facilitates later discussion, particularly in Section 3, where we connect them to complex moduli in the mirror curve, e.g., $(b,b',c,c')=(1,1,R^2,R^2)$ on Page 13. A clarifying footnote has been added on Page 4.

  2. We agree the phrase "ignoring $g_q$" was unclear, and the condition $e^{i k_x}=-b/c' Q^{-1/2}$ does not eliminate $g_q$. Here, we simply set $g_q=0$ by hand so that the spurious term disappears. We have revised the wording accordingly. The missing minus sign and other typos have also been corrected.

  3. In response to the Referee's suggestion, we have added a comment in the Conclusion noting that a quantum version of the Picard-Fuchs equation may be key to a proof of the conjecture, though to our knowledge this difference equation has not yet been explored.

---

## Round 1 · Referee Report · Anonymous (Referee 2) · 2025-4-28

Strengths

-

Report

In this paper the author calculated the generating function of the numbers of closed random walks on either a square lattice or a triangular lattice with a given length and a given signed area, characterized by the number of moves in each hopping direction. For both the square lattice and a special case of the triangular lattice, closed formulae were written down for the generating functions. In the course of the derivation, an interesting connection with exclusion statistics was proposed.

Furthermore, the author proposed a very intriguing connection between the generating function of closed random walks and the quantum A-periods of toric Calabi-Yau threefolds, which is definitely worth further looking into.

I recommend publication of these quite novel results. There are some scientific typos that should be corrected though.

Requested changes

1- On page 6 on the second line above the second equation, $e^{i k_x} = b/c' Q^{-1/2}$ should be corrected to $e^{i k_x} = - b/c' Q^{-1/2}$. 2- On the same page, in the fourth equation, $Z_0$ should be 1 instead of 0.

Recommendation

Ask for minor revision

  • validity: high
  • significance: high
  • originality: high
  • clarity: high
  • formatting: excellent
  • grammar: excellent

Author:  Li Gan  on 2025-05-21  [id 5499]

(in reply to Report 2 on 2025-04-28)

We are grateful to the Referee for the kind remarks regarding the content of our paper. We have carefully reviewed the manuscript and corrected all misprints identified.

---

## Round 2 · Referee Report · Anonymous (Referee 1) · 2025-5-23

Report

Please see attached file.

Attachment

Recommendation

Ask for minor revision

  • validity: -
  • significance: -
  • originality: -
  • clarity: -
  • formatting: -
  • grammar: -

Author:  Li Gan  on 2025-05-26  [id 5522]

(in reply to Report 1 on 2025-05-23)

We appreciate the Referee's feedback and comments. It is reasonable to treat $k_x$ as a free parameter and leave it as it is, since for low enough $N$, the matrix trace $\mathrm{tr}\,H_{\text{sq}}^N$ does not depend on $k_x$ and reproduces the full trace $\mathbf{Tr}\,H_{\text{sq}}^N$. This means that the several lowest cluster coefficients calculated via the $k_x$-dependent partition function are $k_x$-independent. However, the $k_x$-dependent partition function is complicated by the $k_x$-dependent Wilson loop term appearing in the secular determinant $\det(I-zH_{\text{sq}})$.

In our paper, we simplify the secular determinant by eliminating the Wilson loop term. We choose a $k_x$ such that $e^{ik_x} = − b/c' Q^{−1/2}$, ensuring that $f_q=0$, and we also set $k_y=0$ and $g_q = 0$. With these choices, the Wilson loop term vanishes. Note that, in general, the value of $k_x$ that makes $f_q=0$ does not necessarily make $g_q=0$, so we treat $g_q$ as zero, as stated in the paper. Put simply, we not only remove the two corner elements of $H_{\text{sq}}$, but also fix $k_x$ and $k_y$ to eliminate the Wilson loop term, thereby facilitating the subsequent calculation.

The approach we present in the paper is an interesting one, as it allows the calculation of the secular determinant $\det(I-zH_{\text{sq}})$ in the standard exclusion partition function way. Other approaches to computing $\mathrm{tr}\,H_{\text{sq}}^N$ are certainly possible but would require more effort.

---

## Round 2 · Referee Report · Anonymous (Referee 1) · 2025-5-27

Report

The author probably misunderstood my point and did not follow my suggestion for making the presentation more consistent. As this is a minor issue and not worth litigating, I recommend publishing the paper in its present form.

Recommendation

Publish (meets expectations and criteria for this Journal)

---

## Round 2 · Referee Report · Anonymous (Referee 2) · 2025-6-4

Report

The author has addressed all the issues I raised for the previous version. I thus recommend the publication of this paper.

Recommendation

Publish (meets expectations and criteria for this Journal)

---

## Round 2 · Author Response

Dear Editor,

We hereby resubmit our manuscript [https://arxiv.org/abs/2412.21128v2], incorporating revisions that address the Referees' comments as well as the editorial suggestions. We are grateful for the Referees' overall positive assessment of our work, and we appreciate their constructive feedback and valuable suggestions. Below, we provide a brief response to each Referee. A detailed list of changes made to the revised manuscript is included under the "List of changes" section in the submission system.

Response to Referee \#1

We thank the Referee for the helpful comments and suggestions. The Referee correctly pointed out that the parameters introduced to enumerate closed paths with different step counts in each direction are redundant and can be reduced. However, we have chosen to retain all variables to make the step counts in each direction explicit. This choice facilitates further discussion, particularly in Section 3, where we connect these parameters to complex moduli in the mirror curve. We have clarified this decision with an explanatory footnote added on Page 4 of the revised manuscript.

The Referee also noted ambiguity in the phrase ``ignoring $g_q$''. We clarify that the condition $e^{i k_x}=-b/c' Q^{-1/2}$ does not eliminate $g_q$, and to disregard the ``spurious'' term, we simply treat $g_q$ as zero. We have revised the manuscript accordingly to reflect this explanation more clearly. Additionally, all scientific typos noted by the Referee have been corrected.

In response to the Referee's suggestion to add a comment on the proof of the conjecture in Section 3, we have revised the Conclusion section. There, we suggest that a quantum analog of the classical Picard–Fuchs differential equation may be key to a potential proof. However, to the best of our knowledge, such a difference equation has not yet been studied and remains an open problem.

Response to Referee \#2

We are grateful to the Referee for the kind remarks regarding the content of our paper. We have carefully reviewed the manuscript and corrected all misprints identified.

Thank you for your kind attention.

Best regards,
Li

---

## Round 2 · List of Changes

1. Textual revisions

Page 4: Added a footnote addressing Referee #1's suggestion regarding the redundancy of parameters introduced to enumerate closed paths.

Pages 6, 8: Replaced ignore $g_q$'' withneglect $g_q$ by treating it as zero''.

Page 14: Revised the explanation regarding the quantum counterpart of the classical Picard–Fuchs differential equation; added acknowledgments to the anonymous Referees for their insightful comments and suggestions.

  1. Corrections of misprints

Page 4: this Hamiltonian $H$ describes the Hofstadter model on a triangular lattice'' $\rightarrow$the Hamiltonian $H_{\text{tri}}$ describes the Hofstadter model on a triangular lattice''.

Pages 6, 8: $e^{i k_x}=b/c' Q^{-1/2}$ $\rightarrow$ $e^{i k_x}=-b/c' Q^{-1/2}$

Page 6: $Z_0=0$ $\rightarrow$ $Z_0=1$

---

## Editorial Decision

published